# Presence and abundance of bacteria with the Type VI secretion system in a coastal environment and in the global oceans

**Michael W. Kempnich, Marilou P. Sison-Mangus** *

Department of Ocean Sciences and Institute of Marine Sciences, University of California, Santa Cruz, Santa Cruz, California, United States of America

* msisonma@ucsc.edu

## Abstract

Marine bacteria employ various strategies to maintain their competitive advantage over others in a mixed community. The use of Type VI Secretion Systems (T6SS), a protein secretion apparatus used as a molecular weapon for interbacterial competition and eukaryotic interactions, is one of the competitive strategies that is least studied among heterotrophic bacteria living in the water column. To get an insight into the temporal and spatial distribution of bacteria with T6SS in this portion of the marine environment, we examine the presence and abundance of T6SS-bearing bacteria at both local and global scales through the use of metagenome data from water samples obtained from the coast of Monterey Bay and the TARA Oceans project. We also track the abundance of T6SS-harboring bacteria through a two-year time series of weekly water samples in the same coastal site to examine the environmental factors that may drive their presence and abundance. Among the twenty-one T6SS-bearing bacterial genera examined, we found several genera assume a particle-attached lifestyle, with only a few genera having a free-living lifestyle. The abundance of T6SS-harboring bacteria in both niches negatively correlates with the abundance of autotrophs. Globally, we found that T6SS genes are much more abundant in areas with low biological productivity. Our data suggest that T6SS-harboring bacteria tend to be abundant spatially and temporally when organic resources are limited. This ecological study agrees with the patterns observed from several in vitro studies; that T6SS could be an adaptive strategy employed by heterotrophic bacteria to obtain nutrients or reduce competition when resources are in limited quantity.

## Introduction

Marine bacteria constitute the largest portion of biomass in the world's oceans [1, 2] and are responsible for the cycling and regeneration of nutrients throughout the ocean [3]. Most regenerated primary production relies on nutrients that pass through the microbial loop, whereby bacteria and other microorganisms metabolize organic compounds and return them to non-organic, bioavailable forms [4]. Function and efficiency within the microbial loop are

**Data Availability Statement:** All relevant data are within the manuscript and its Supporting Information files.

**Funding:** Unfunded study.

**Competing interests:** The authors have declared that no competing interests exist.

affected by the composition and activities of the microbial community. For example, diverse functional groups of bacteria contribute to nutrient cycling, carbon transformation, and ecosystem functions [5, 6]. Fenchel [3] describes functional categories of bacteria acting within the microbial loop in very specific metabolic processes, such as the oxidation of ammonia or C-1 carbon compounds. Shifts in bacterial community composition could affect functional group representation and may affect the efficiency of these metabolic pathways.

Bacteria in ocean ecosystems are generally considered to be concentrated between $10^3$ cells/mL—$10^6$ cells/mL, which is thought to be much lower than their carrying capacity [7, 8]. This difference is due in part to predation on bacteria, both by larger organisms and by other bacteria [9]. Interbacterial competition takes many forms, some involving the killing of neighboring bacterial cells and uptake of lysed nutrients [10]. Evidence of interbacterial competition have been reported in laboratory settings [10–12], in the human gut [13], among plant-associated microbiota [14], in aquatic zooplankton-associated microbiota [15], and are known to be facilitated by the use of the Type VI secretion system (T6SS).

T6SS is one of the secretion systems that is hypothesized to mediate bacterial killing and niche occupancy [16–18]. It is present in up to 25% of Gram-negative bacteria, mostly within the Proteobacteria [19–22]. This secretion system is a versatile molecular machine that delivers effector proteins directly into target cells and is both a factor in pathogen virulence and a mechanism for killing other bacteria in laboratory studies [17, 23]. Bacteria with T6SS display contact-dependent delivery of anti-microbials and evidences suggest that they have evolved this mechanism for use as a competitive and predatory means, which was later co-opted as a mechanism for pathogen virulence [16, 17, 24]. The structure of T6SS is strikingly similar to a bacteriophage tail, with a transmembrane anchor and contractile sheath, which extends to puncture target cells [21, 25]. After puncturing a target cell, T6SS delivers effector proteins that are toxic to the target cell. These effectors can attack the cell wall, cell membrane, or nucleic acids depending on the specific protein being delivered, leading to cell lysis [26]. The primary extracellular structure of T6SS is comprised of two proteins, valine-glycine repeat G (*VgrG*) and haemolysin coregulated protein (*Hcp*), which are used as proxies for T6SS activity [16, 25, 27]. *VgrG* forms the "cap" of the T6SS assembly and is propelled by a sheath formed from the *Hcp* structural protein; both proteins and several others are needed for T6SS to function [12, 25, 28]. While the structure of T6SS is well conserved among T6SS containing bacteria, the gene corresponding to the structural proteins is not. To date, most T6SS pathways have been identified by protein structure rather than by gene sequence [11, 21, 27, 28].

T6SS is one of the means of interbacterial competition found widely amongst sequenced bacteria from various environment, including bacteria from the marine environment such as coral reefs [22, 29]. Currently, knowledge is limited about interbacterial competition using T6SS among heterotrophic bacteria that live in the marine water column where particulate and dissolved organic carbon are the primary food sources. What environmental or biological determinants might drive T6SS usage among these bacteria? Bacterial predation and competition play a role in modulating the bacterial community [10], but this process and its implications are poorly understood compared to competitive pressures in eukaryotic communities. In this study, we investigated the presence and abundance of bacteria with T6SS genes at a coastal site in Monterey Bay, California, and have extended the survey globally using the dataset from the Tara Oceans Project. We also assessed the environmental characteristics that may drive their abundance using multivariate correlational analyses. We report that T6SS-harboring bacteria are predominantly found to be particle-associated and that they changed in abundance with time and space with specific environmental conditions. Our study provides insight into the ecology of bacterial populations with T6SS in the marine pelagic environment and the putative importance of T6SS for understanding changes in the bacterial community.

## Materials and methods

### Water sampling

Bacterial community sampling was done weekly at Santa Cruz (SC) Wharf, an ocean monitoring site for harmful algal bloom in Monterey Bay, Central California (CENCOOS). Sampling times and location corresponded with sampling for the SCCOOS HAB database [30], the source for the environmental and biological metadata examined in this study. Water samples were collected at the end of the SC Wharf, approximately 700 meters from the shore, with a maximum depth of approximately 10 meters. Geographical coordinates for this sampling site are approximately 36˚57'28.6"N 122˚01'04.1"W. SC Wharf is open to the public and requires no permits for the sampling activities conducted for this study.

A Niskin bottle was used to collect three water samples of 1.3 liters each at 0, 1.5, and 3 meters below the ocean surface. These samples were homogenized, then divided into three replicates of 1 liter. Each replicate was filtered first through sterile, prepackaged 3.0 μm cellulose nitrate membrane filters (Sartorius, USA) and then through sterile, autoclaved 0.2 μm Durapore membrane filters (Millipore, USA) on 47mm Nalgene filter towers. Immediately after filtering, filters were stored at -80˚C until DNA extraction. These filter sizes represent the particle-attached bacterial community (3.0 μm), which adheres to particles in the water and is trapped on the larger filter size, and the smaller filter size sequentially traps the free-living bacteria (0.2 μm) which pass through the larger filter size.

### DNA extraction, sequencing and analysis

Whole community DNA was extracted from both 3.0 μm and 0.2 μm filters using the Power-Water DNA Isolation kit (MO BIO Laboratories, Inc., CA), per manufacturer instructions. Briefly, samples were physically lysed and removed from filters by bead-beating, then chemically lysed using the proprietary chemicals from the PowerWater kit. Following lysis and removal of debris, DNA was captured on the provided filter column, washed, and eluted into sterile vials. DNA extract was tested for quality and quantity and frozen at -80˚C before sequencing.

DNA samples were tested for DNA quality through PCR amplification using universal 16S rRNA primers 515F (5'-GTGCCAGCMGCCGCGGTAA-3') and 806R (5'-GGAC-TACHVGGGTWTCTAAT-3') that target the V4 region [31]. Samples with no amplification were tested again following a 1:10 dilution in nanopure PCR water, after which the remainder of un-amplified samples were removed from further analysis. Good quality DNA extracts were quantified using the Quant-iT PicoGreen dsDNA assay (Molecular Probes, Inc. #P7589). Extract aliquots were diluted 50x in 1x TE buffer and PicoGreen dye. DNA standards were created ranging from 0–2000 ng/mL. DNA samples and standards were read on a Spectramax M2e plate reader spectrophotometer (Molecular Devices) at Ex = 480nm and Em = 520nm. Fluorescence readings were converted to DNA concentration using the calculated linear standard curve ($R^2 > 0.99$). Quantified DNA aliquots normalized to concentrations between 1-10ng DNA/μL were sent to the Sequencing Facility at Argonne National Laboratory for Illumina MiSeq Next-Generation Sequencing.

**Metagenome sequencing and analysis.** Ten samples from five sampling dates were chosen for metagenome sequencing from the SC Wharf sample set. These samples were selected to represent ranges of seasons, environmental conditions of high and low chlorophyll, including toxic phytoplankton bloom (*Pseudo-nitzschia*, $1.23 \times 10^5$ cells/ml), and putative abundance of bacterial genera with T6SS. Metagenome samples were checked for DNA quality before library preparation and sequencing using Illumina HiSeq by GENEWIZ. Paired-end 150 bp sequences

(454,071,682 total sequences, mean quality score = 38.8) were used in building BLAST databases using BLAST+ 2.9.0 and searched for homologous *hcp*, *vgrg*, *clpV*, *tssA*, and *tssB* sequences belonging to known bacterial genera with T6SS (see S1 Table for NCBI and UniProtKB accession numbers). Sequence matches were filtered with an E-value cut-off of 1 x $10^{-4}$.

Similarly, fifty-two metagenome samples available in the TARA Oceans database were downloaded (https://www.ebi.ac.uk/metagenomics/studies/ERP001736) and data-mined for known bacteria with T6SS to determine their abundance pattern worldwide. BLAST databases were built from the complete set of metagenome sequence reads and were searched for homologous sequences of *hcp* and *vgrg* sequences belonging to known bacterial genera with T6SS, with filter cut-off as above. Matching gene counts were converted to reads per million (rpm) mapped reads. RPM was calculated as the total sequence reads for each identified bacterial genus with T6SS.

**16s rRNA sequence processing.** Post-processing of sequences from 331 samples from 55 sampling time points was carried out using the Quantitative Insights Into Microbial Ecology (QIIME 1.9.1) pipeline [32] as laid out in Sison-Mangus et al., 2014 [33]. Briefly, OTUs were picked using Pynast with 97% similarity [31], then taxonomically assigned using RDP classifier 2.2 [34] and BLAST [35]. Singletons, mitochondria, and chloroplast sequences were removed before data analysis. Raw sequencing reads have been deposited in Genbank under BioProject ID PRJNA648155 with respective BioSample IDs in S2 Table.

Following OTU taxonomic assignment, a compositional data analysis (CoDA) framework [36] was used following the methods laid out in Quinn et al. [37]. The OTU table was trimmed down to OTUs with at least 50 representative sequences among all samples, leaving 1274 OTUs in the 0.2 um size fraction (out of 27,780 OTUs) and 1977 OTUs in the 3.0 um size fraction (out of 39,823). Using cmultRepl from the zCompositions R package [38], zeros in the dataset were transformed to small positive values using the Bayesian multiplicative approach strategy [39], while maintaining ratios for non-zero counts followed by converting the raw counts to abundance table. The data were then log-transformed into centered log-ratio (clr) values using zCompositions R package to determine the abundance of OTUs relative to the per-sample average (geometric mean). T6SS-harboring bacterial OTUs were filtered from the clr abundance table and were summed for each genus for downstream analysis. Bacterial genera with known T6SS were chosen as those containing five core structural genes of the T6SS (see S1 Table) previously identified and confirmed in the SC Wharf metagenome sequences. In most cases, bacterial genera with T6SS were represented by multiple OTUs within this data set. R scripts specifics for this step is found in (https://github.com/M-Kempnich/Presence-and-abundance-of-T6SS/blob/master/CLR%20Transformation.R).

## Statistical analyses

The lifestyle of T6SS-bearing bacteria (free-living, particle-attached, or mixed) was examined by comparing abundances of bacterial genera between size fractions using the ALDEx2 R package [40]. Raw count abundances of T6SS-harboring bacteria were subjected to centered-log-ratio transformation, then 128 Monte Carlo simulations were run to estimate within- and between-treatment differences (0.2 μm filter fraction against 3.0 μm filter fraction). Each bacterial genus was compared pairwise in its abundance between the 3.0 μm filter fraction and the 0.2 μm filter fraction using Welch's t-test, with p-values corrected for multiple testing via the Benjamini-Hochberg method. Genera showing a statistically significant higher abundance in one of the size fractions was assigned to that corresponding lifestyle.

To elucidate the relationships between T6SS-bearing bacterial abundance and the environmental variables that may potentially drive them, the correlations were examined using

Redundancy Analysis (RDA), a direct gradient analysis technique that combines and summarizes the linear relationships of multiple explanatory variables (i.e., environmental variables) via constrained ordination of a set of response variables (i.e., T6SS-bearing bacterial populations). RDA was executed in the VEGAN 2.5–6 package [41] in R 3.6.2 using clr transformed values of bacterial abundance. Environmental variables include temperature ($^o$C), chlorophyll level (mg/ml), nutrients (nitrate, phosphate, silicate, ammonia, μM), and toxic *Pseudo-nitzschia* (Pn) diatom cells (cells/ml), which were highly abundant during the sampling time frame. S3 Table contains the value ranges of these variables. Variance inflation factor (VIF) was computed to check the collinearity among environmental variables, with VIF > 10 excluded from subsequent analysis. Anova.cca (R-package Vegan) function was used to test the statistical significance of the environmental variables that explained the abundance of T6SS-bearing bacterial populations, using 999 Monte Carlo permutations and Benjamini and Hochberg multiple test correction of p-values (< 0.05). The biplot generated shows the length and angle of the vectors indicating the importance of these environmental variables. All R codes used are available at https://github.com/M-Kempnich/Presence-and-abundance-of-T6SS.

RDA was also used in analyzing data from the TARA Oceans to determine the environmental drivers of T6SS-bearing bacteria at a global scale, with bacterial abundance represented by *vgrg* and *hcp* gene rpm mapped reads as a set of response variables. Twenty-nine explanatory variables (see S4 Table) were initially considered for the RDA analysis; variables with VIF <10 was excluded and forward selection with 999 permutation tests was used to test the contribution of each variable on all axes. The overall RDA model significantly improved when six variables were used: number of autotroph cells (cells/ml), nitrite-nitrate ($NO_2$+$NO_3$, μM), oxygen concentration ($O_2$, μmol/kg), bacterial functional richness (Tax. Rich.) and water sample depth (Depth, meters). Significance testing of the variables was carried out using permutations and multiple test correction of p-values as mentioned above.

## Results

### Abundance of bacteria with T6SS genes in the coastal ocean

Metagenome sequence analysis from SC Wharf revealed the presence of core T6SS genes belonging to twenty-one genera of bacteria across ten samples from five time points (Fig 1). Metagenome samples were chosen from three seasons and broad ranges of environmental conditions such as chlorophyll level, phytoplankton bloom, and nutrient levels. Sequences identified as *vgrG*, *hcp*, *clpV*, *tssA*, and *tssB* (core structural genes of T6SS, S1 Table) were found to be present in both 3.0 μm and 0.2 μm water fractionated samples at varying abundances, indicating the presence of these T6SS-bearing bacteria in our samples. These 21 genera belonging to classes Planctomycetia (*Planctomyces*), Alpha-proteobacteria (*Bradyrhizobium*, *Mesorhizobium*, *Methylobacterium*, *Agrobacterium*, *Ruegeria*, *Rhodobacter*, *Sphingomonas*), Beta-proteobacteria (*Janthinobacterium*), Epsilon-bacteria (*Arcobacter*), and Gamma-Proteobacteria (*Acinetobacter*, *Enterobacter*, *Francisella*, *Halomonas*, *Pseudoalteromonas*, *Pseudomonas*, *Psychromonas*, *Serratia*, *Shewanella*, *Teredinibacter*, *Vibrio*) made up the T6SS-harboring bacterial populations that were tracked in our subsequent study. T6SS sequence reads were less abundant (4.2–5.5 rpm) in April and July of 2014 but were more abundant in December 2014 (15–19 rpm) and were dominated by *Halomonas*, *Pseudomonas*, *Serratia*, *Methylobacterium*, *Ruegeria*, *and Arcobacter* in both water size fractions.

Guided by the metagenome results, we assessed the abundances of these T6SS-bearing bacteria through time in our local time-series coastal samples using 16S rRNA data. The twenty-one bacterial genera with putative T6SS were found to be consistently present at varying

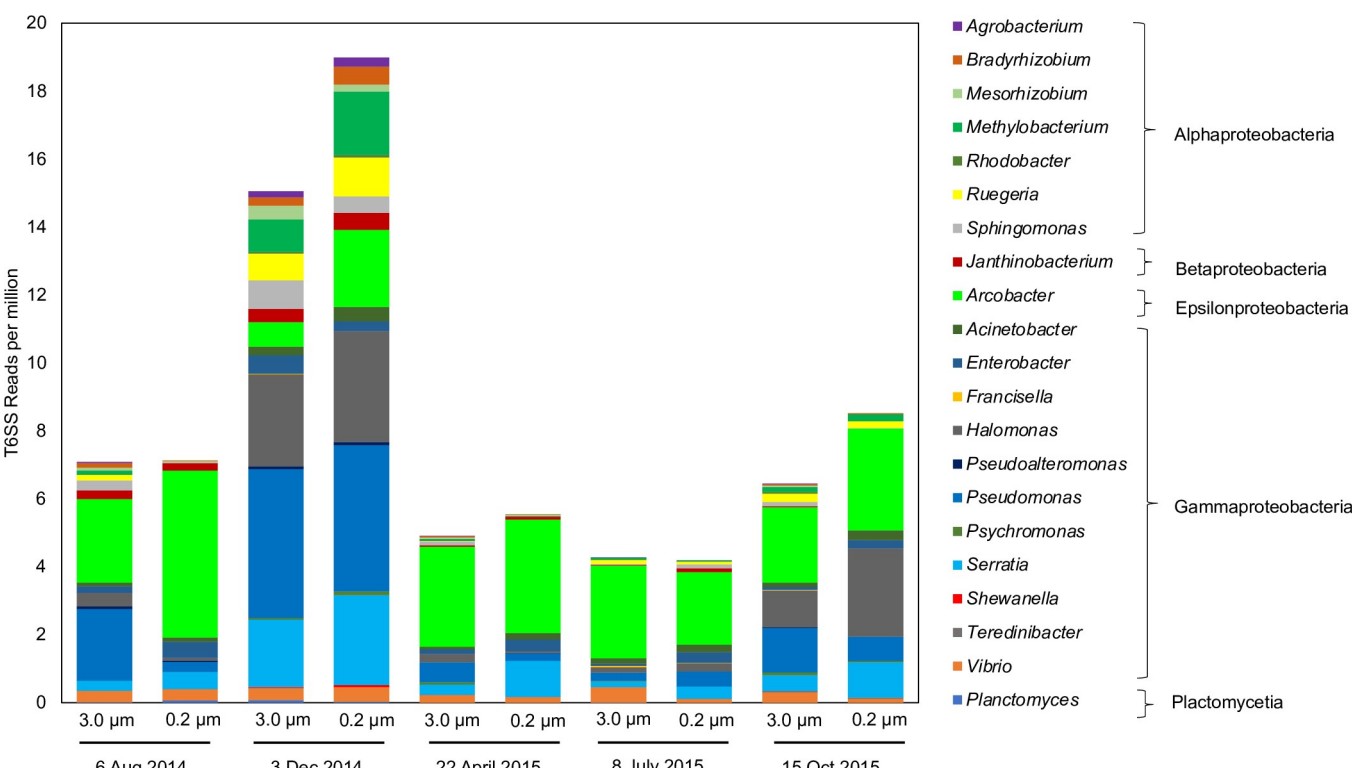

**Fig 1. Metagenome sequence reads (in RPM) of T6SS core genes from 3.0 and 0.2 μm water size fractions by sample date.** Samples span seasons and broad ranges of environmental conditions such as high and low chlorophyll level, toxic phytoplankton bloom, and nutrient levels. Twenty-one bacterial genera from several classes known to harbor five core T6SS genes were confirmed and are present at varying rpm across samples.

abundances through time in both 3.0 μm and 0.2 μm water fractions (Fig 2). Fifteen genera were present in every analyzed sample for both water fraction, while six bacteria (*Agrobacterium*, *Bradyrhizobium*, *Mesorhizobium*, *Methylobacterium*, *Planctomyces*, *Serratia*) where only found in the particulate-attached fraction. Bacteria with T6SS ranged in relative abundance from 1% to 11% of the total bacterial community in these samples. The changes in the abundance of T6SS-bearing bacteria relative to the per-sample average was plotted as a heatmap profile using clr transformed values (Fig 2). The majority of the T6SS-bearing bacteria tended to have increased abundance at SC Wharf in colder months, and lower abundance at the start of summer with a minimum in late June and early July, a similar pattern seen in the T6SS metagenome survey (Fig 1). This seasonality appeared in data across 2014 and 2015 and both water fractions, aside from one high abundance outlier in August of 2014, dominated by *Arcobacter*.

**Diverse T6SS-bearing bacteria occupy different niches.** Because we observed marked differences in the relative abundances of each bacterial genera between water size fractions, we further assessed whether T6SS-bearing bacteria are much more predominant in particulate-attached fraction, free-living fraction, or they have a mixed lifestyle. Each genera's abundances between size fractions were compared across all samples using the ALDEx2 R package followed by statistical significance test using Welch's t-test (Table 1). This analysis showed that four genera are preferentially free-living (*Acinetobacter*, *Francisella*, *Pseudomonas*, and *Ruegeria*) ($p < 0.05$), while four genera have mainly particle-attached lifestyle (*Pseudoalteromonas*, *Psychromonas*, *Shewanella*, and *Vibrio*) ($p < 0.05$). Six genera (*Agrobacterium*, *Bradyrhizobium*, *Mesorhizobium*, *Methylobacterium*, *Planctomyces*, and *Serratia*) were observed solely in the 3.0 μm size fraction and are therefore considered particle-attached. The remaining seven

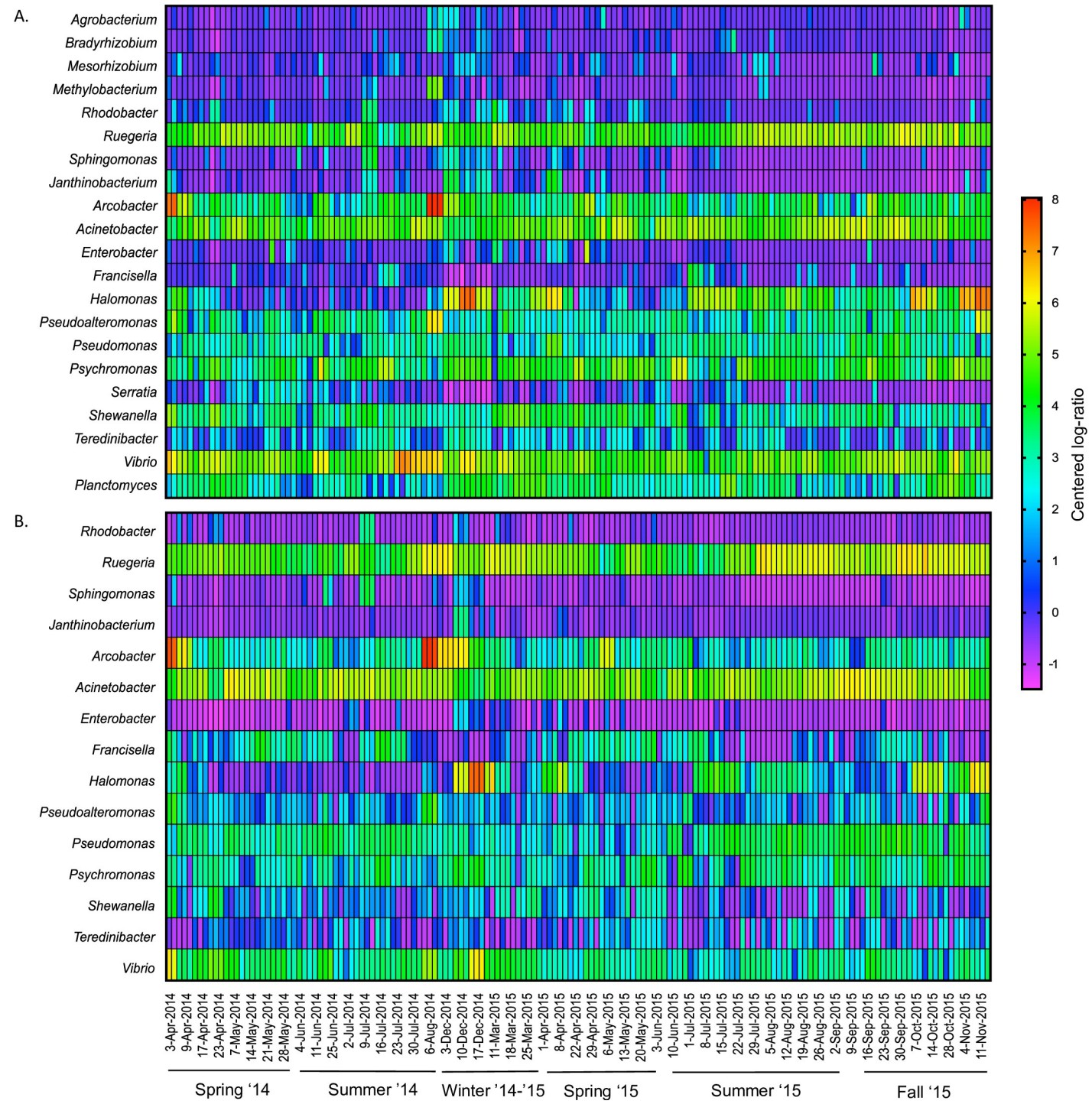

**Fig 2. Heatmap profile showing the abundance of various T6SS-bearing bacterial genera from 2014–2015 samples.** (A) Particle-attached (3.0 μm) filter size fraction. (B) Free-living (0.2 μm) filter size fraction. OTU abundance relative to per-sample geometric mean is based on clr transformed values of sequence reads belonging to each bacterial genus across duplicate or triplicate samples.

genera (*Arcobacter*, *Enterobacter*, *Halomonas*, *Janthinobacterium*, *Rhodobacter*, *Sphingomonas*, and *Teredinibacter*) did not show a statistically significant difference in percent abundance between water fractions and have been labeled as having a "mixed" lifestyle.

**Table 1. Differential abundance of T6SS-bearing bacteria between filter sizes.**

| T6SS-bearing Bacterial Genera | Effect size | Difference (between) | Difference (within) | Expected Benjamini Hochberg P-value | Lifestyle[A] |
|---|---|---|---|---|---|
| *Planctomyces* | * | * | * | * | Particle-attached |
| *Bradyrhizobium* | * | * | * | * | Particle-attached |
| *Mesorhizobium* | * | * | * | * | Particle-attached |
| *Methylobacterium* | * | * | * | * | Particle-attached |
| *Agrobacterium* | * | * | * | * | Particle-attached |
| *Rhodobacter* | 0.0784 | 0.4015 | 4.3120 | 0.4068 | Mixed |
| *Ruegeria* | -0.5689 | -1.2455 | 1.9838 | ***0.0000*** | Free-living |
| *Sphingomonas* | 0.0373 | 0.1974 | 4.1989 | 0.5216 | Mixed |
| *Janthinobacterium* | 0.1223 | 0.5893 | 4.3187 | 0.2768 | Mixed |
| *Arcobacter* | 0.0285 | 0.0733 | 2.3119 | 0.6939 | Mixed |
| *Acinetobacter* | -0.6636 | -1.4238 | 1.9860 | ***0.0000*** | Free-living |
| *Enterobacter* | 0.0129 | 0.0662 | 4.1205 | 0.5492 | Mixed |
| *Francisella* | -0.6784 | -3.5610 | 4.7138 | ***0.0000*** | Free-living |
| *Halomonas* | 0.1268 | 0.7546 | 5.4423 | 0.1732 | Mixed |
| *Pseudoalteromonas* | 0.3682 | 1.0170 | 2.3826 | ***0.0000*** | Particle-attached |
| *Pseudomonas* | -0.5277 | -1.3857 | 2.2430 | ***0.0000*** | Free-living |
| *Psychromonas* | 0.2796 | 0.6769 | 2.1074 | ***0.0003*** | Particle-attached |
| *Serratia* | * | * | * | * | Particle-attached |
| *Shewanella* | 0.5670 | 1.8199 | 2.8282 | ***0.0000*** | Particle-attached |
| *Teredinibacter* | 0.0295 | 0.1291 | 3.4528 | 0.6147 | Mixed |
| *Vibrio* | 0.7008 | 1.4639 | 1.8710 | ***0.0000*** | Particle-attached |

[A]—Calculations performed using aldex from R package ALDEx2, utilizing clr transformation. Effect size is based on 128 Monte Carlo simulations. P-values are based on Welch's t-test performed across Monte Carlo simulations, corrected via Benjamini-Hochberg method for multiple comparisons. Negative effect size indicates higher abundance in the 0.2 μm size fraction, while positive effect size indicates higher abundance in the 3.0 μm size fraction. Lifestyle was assigned based on both effect size and p-value. "Particle-attached" genera show significantly higher abundance in the 3.0 μm size fraction, while "Free-living" genera are significantly more abundant in the 0.2 μm size fraction. "Mixed" bacteria were not significantly different in abundance in either size fraction.

*—indicates genera that were only observed in the 3.0 μm size fraction, and therefore could not be compared via this test.

*Abundance patterns of bacteria with T6SS on a global scale.* Assessment of metagenome data from the TARA Oceans project showed that core T6SS genes of the bacterial genera analyzed in our local study were also found to be universally present in globally distributed samples. BLAST searches of TARA Oceans metagenomes returned reads of *vgrg* and *hcp* from every tested sample, ranging from 0.002 to 1.1 reads per million mapped reads (RPM) corresponding to eight bacterial genera commonly found in samples from SC Wharf. RPM of T6SS genes at each sampling site can be found in Fig 3. T6SS genes were most common in samples from the southern Atlantic Ocean gyres and the Indian Ocean gyres and were least abundant in samples taken near landmasses.

*Relationship between T6SS-bearing bacterial abundance and environmental factors.* To get a better understanding of the environmental factors that could drive the abundance of T6SS-bearing bacteria, RDA was used to assess the associations of bacterial abundance and several physico-chemical-biological environmental parameters. Arrow length in the ordination plots represents the strength of each environmental parameter's relationship to T6SS-bearing bacterial composition.

In the SC Wharf time-series samples (Fig 4), The RDA model testing the effects of environmental variables on the abundance of T6SS-bearing bacteria were significant for both 3.0 and 0.2 μm water fractions (p< 0.001). In the particle-attached fraction (Fig 4A), the model

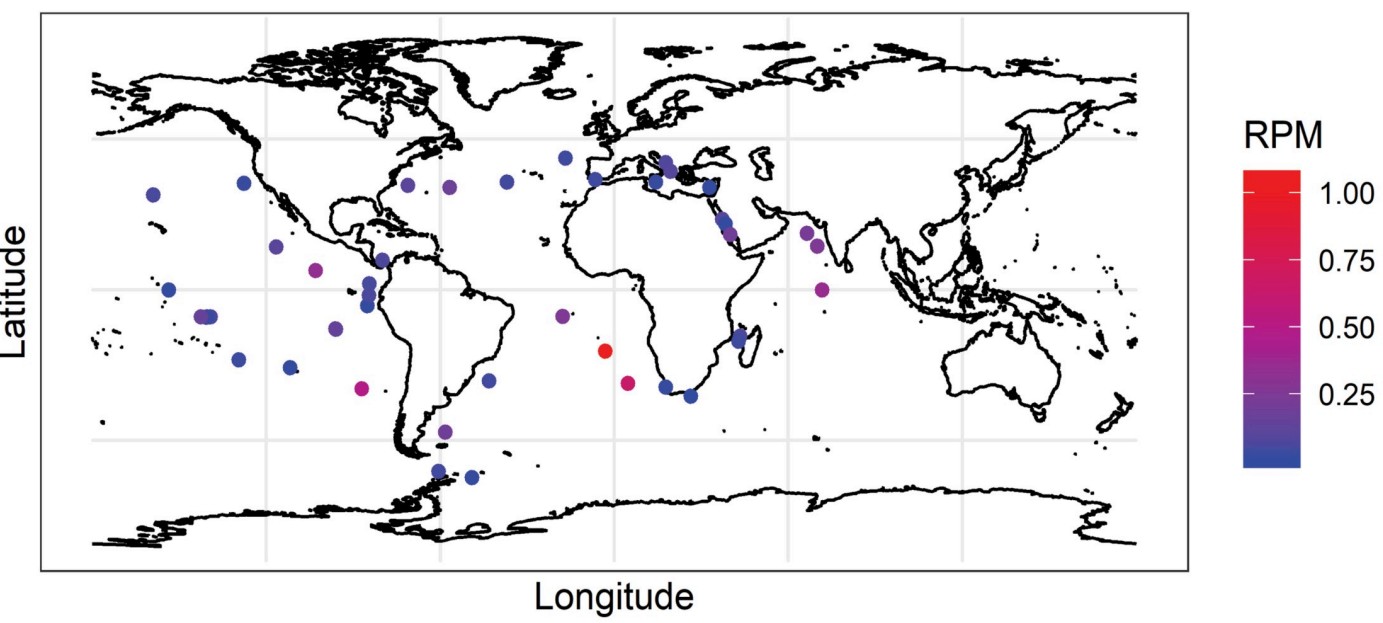

**Fig 3. Heatmap of the global distribution of T6SS *hcp* and *vgrg* gene sequences.** Reads per million (rpm) mapped reads taken from TARA Oceans metagenome dataset. Hotspots of T6SS abundance appear in the South Atlantic and Indian Ocean gyres, known areas with extremely low biomass and low nutrient concentration. Reads matching known *hcp* and *vgrg* gene sequences were acquired using BLAST+ 2.9.0. Map was built using data from R Natural Earth. Code is available at https://github.com/M-Kempnich/Presence-and-abundance-of-T6SS.

explained 22.8% of the observed total variation, with the first and second axes accounting for 38.0% and 33.1% of the constrained variability in the T6SS-harboring bacterial population, respectively. The first four RDA axes were significant (p< 0.008), explaining 93% of the variation, suggesting that the represented variations are more structured than random. All environmental variables examined were statistically significant drivers of T6SS-bearing bacterial genera (Benjamini and Hochberg multiple test correction, p<0.01). We observed three distinct patterns of association between T6SS-bearing bacteria and environmental variables. The nutrient variables, for instance, were highly associated with the majority of the T6SS-bearing bacteria. Silicate was a strong predictor of T6SS- bearing bacterial abundance (F = 12.43, p = 0.001), and was positively associated with the bacterial genera *Janthinobacterium*, *Sphingomonas*, *and Rhodobacter*. Phosphate (F = 4.71, p = 0.001), nitrate (F = 2.88, p = 0.006), and ammonia (F = 7.75, p = 0.001) shared a similar space on RDA1 and RDA2 axes with silicate. Phosphate, in particular, correlates with *Serratia* and *Shewanella populations*. Notably, the abundance of T6SS-bearing bacteria influenced by nutrients, has shown no association with chlorophyll or toxic *Pseudo-nitzschia* cell density. Moreover, many bacterial genera responded negatively to the combination of toxic diatom (F = 2.59, p = 0.009) and chlorophyll concentration (F = 5.67, p = 0.001), such as *Halomonas*, *Pseudomonas*, *Planctomyces*, *Agrobacterium*, and *Bradyrhizobium*. Temperature is also a significant environmental driver (F = 6.54, p<0.001), specifically for the bacterial genera, *Ruegeria*, *Acinetobacter*, and *Francisella*.

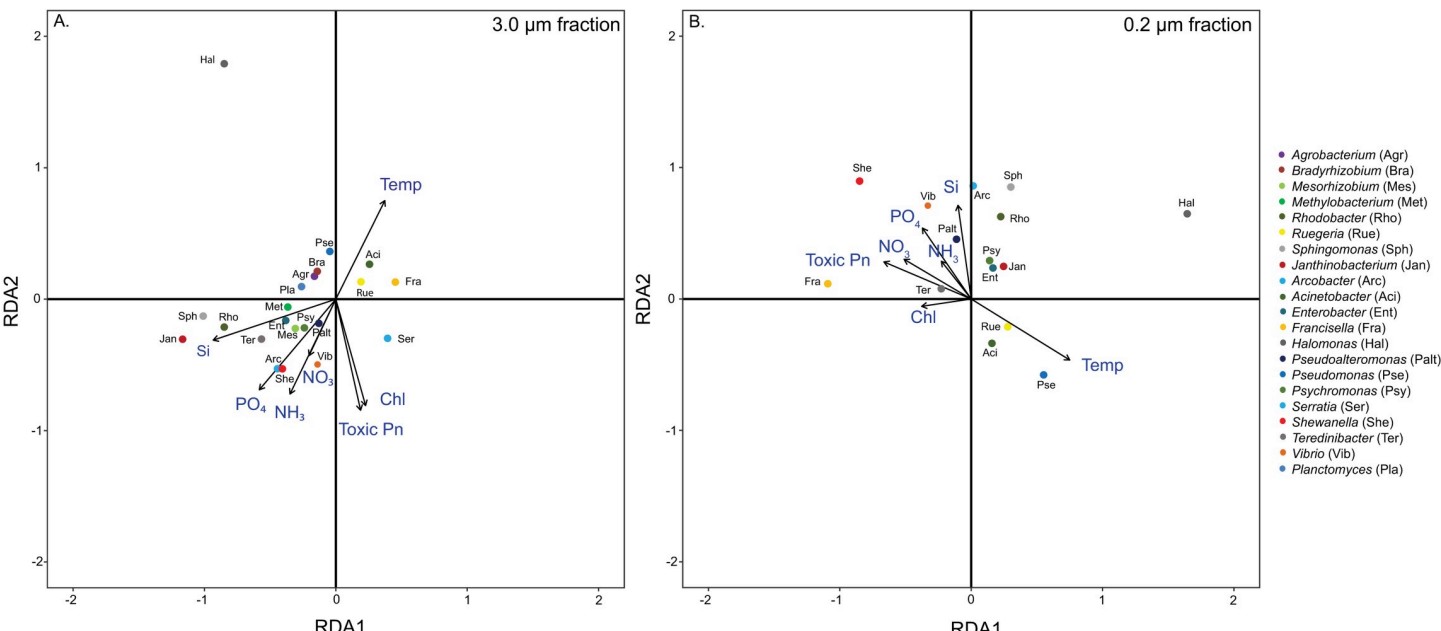

**Fig 4. Redundancy analysis (RDA) demonstrating the association between abundance of T6SS-bearing bacteria and environmental characteristics from SC Wharf time series data.** (A) Relationships of particle- attached (3.0 μm water fraction) community and (B) free-living (0.22 μm water fraction) community with physico-chemical-biological parameters. Considered environmental factors were toxic *Pseudo-nitzschia* (Toxic Pn, cells/mL, nitrate (NO$_3$, μM), ammonia (NH$_3$, μM), phosphate (PO$_4$, μM), silicate (Si, μM) concentrations, chlorophyll concentration (chl, mg/m$^3$), and water temperature (Temp, ˚C)). Arrow length represents strength of the relationship of each environmental parameter to the composition of T6SS-bearing bacteria. Colored circles represent bacterial genera with T6SS.

In the free-living fraction (Fig 4B), the model explained 22.2% of the observed total variation. The first three RDA axes were significant (p<0.004, 75.8% combined), with the first two axes explaining 41.2% and 21.5% of the constrained variation in T6SS-harboring bacterial abundance. As in the particle-attached fraction, all environmental variables examined have VIF < 10 and were statistically significant drivers of the examined bacterial genera (p<0.01). Notably, most bacterial genera showed no association with chlorophyll, while *Halomonas*, *Sphingomonas*, *Acinetobacter*, *Janthinobacterium*, *Psychromonas*, and *Enterobacter* responded negatively to chlorophyll concentration (F = 3.71, p = 0.002). *Ruegeria*, *Pseudomonas*, and *Acinetobacter* responded to increased temperature (F = 8.60, p = 0.001). Nutrients including silicate (F = 13.19, p = 0.001), phosphate (F = 5.43, p = 0.001), nitrate (F = 5.43, p = 0.001), and ammonia (F = 2.77, p = 0.015) grouped together, and *Shewanella*, *Vibrio*, and *Pseudoalteromonas* responded to some combination of these factors.

The TARA Oceans metagenomes show similar patterns with those observed in the SC Wharf samples (Fig 5). Among twenty-nine explanatory variables (S4 Table), 19 variables were selected with a VIF<10. RDA with forward selection was used to identify the variables contributing to the variation in T6SS-bearing bacterial population. Eight bacterial genera and the total T6SS reads per sample were used as response variables for this analysis. The RDA model testing the effect of several environmental drivers on the population abundance of T6SS-bearing bacteria on a global scale was significant (p<0.045). The model explained 31.6% of the total variation, with 97.6% and 2.2% contribution of the first and second axes, respectively. Only water depth showed significant contribution to the variation in abundance of T6SS-bearing bacterial population (F = 11.4, p = 0.002), and this environmental factor was tightly associated with the abundance of total T6SS reads per sample and the *Pseudo-alteromonas* population. Many of the nutrients have strong collinearity (VIF >10), and only silicate and nitrite-nitrate

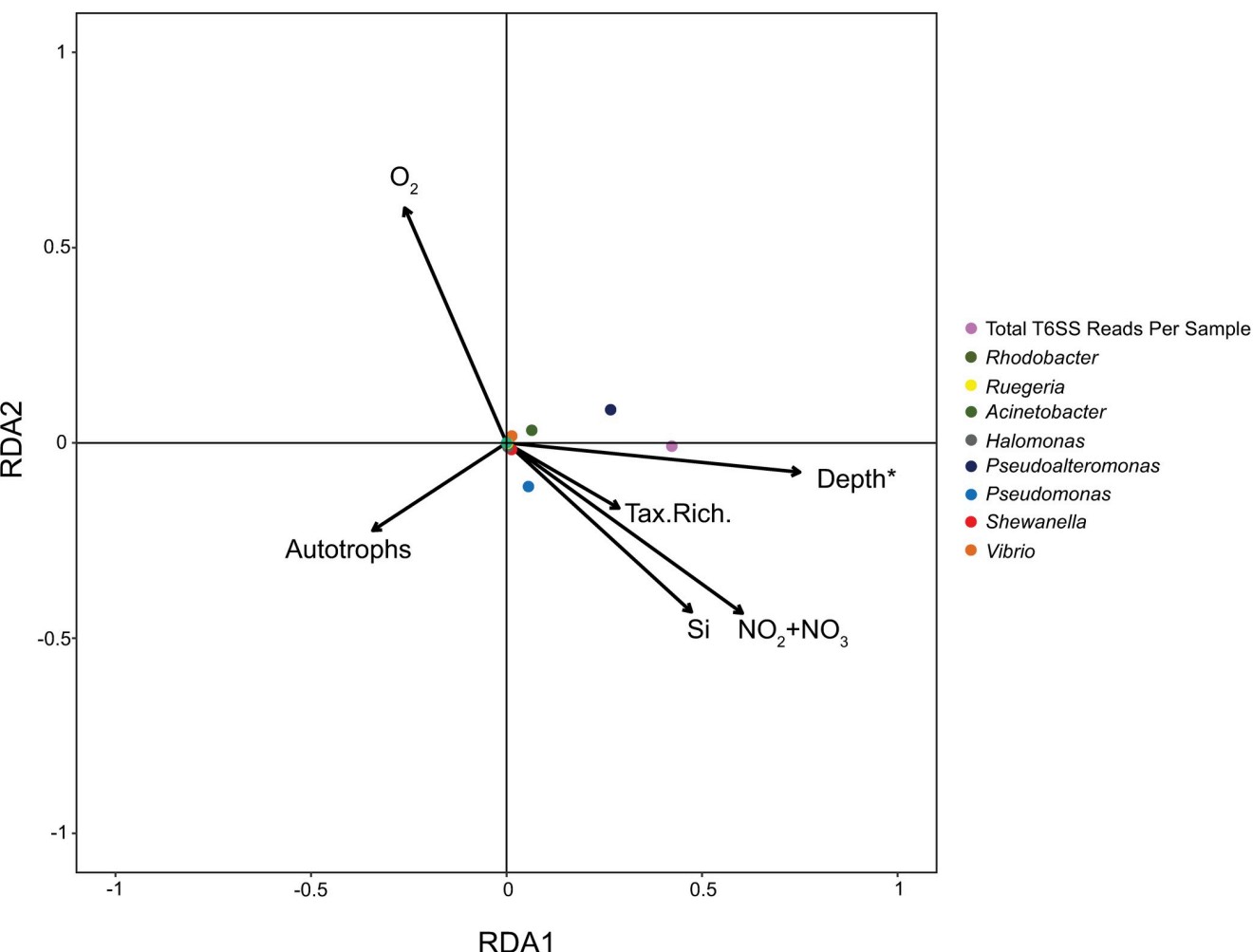

**Fig 5. RDA analysis showing the association of T6SS reads (in rpm) and environmental variables from the TARA Oceans metagenome samples.**
Colored circles represent T6SS *vgrG* and *hcp* gene reads from eight bacterial genera. Six environmental variables that contributed to the variation of the global T6SS reads were plotted and these include number of autotroph cells (cells/ml), nitrite-nitrate ($NO_2+NO_3$, μM), oxygen concentration ($O_2$, μmol/kg), bacterial functional richness (Tax. Rich.) and water sample depth (Depth, meters). Arrow length represents strength of the relationship of each environmental parameter to the composition of T6SS-bearing bacteria. * indicates significant environmental variable ($p < 0.005$).

were included in the analysis. However, these nutrients did not show significant associations with T6SS-bearing bacterial populations ($p > 0.05$). Moreover, these nutrients tend to increase with depth, which overwhelms latitudinal variation in concentration. All of the T6SS-bearing bacteria examined from the TARA Oceans dataset showed no positive association with autotroph abundance (see Fig 5). Interestingly, the bacterial genera *Acinetobacter*, *Pseudoalteromonas*, and the total T6SS gene abundance, instead appeared to be negatively associated with autotroph abundance, but this relationship did not meet the threshold for statistical significance ($F = 3.01$, $p = 0.06$).

## Discussion

We investigated the presence and abundance of bacteria with T6SS in a local coastal environment and the distribution of these populations in the global ocean to understand their

population dynamics and the environmental characteristics that may trigger the use of their T6SS. Our study showed that T6SS-bearing bacteria are omnipresent in the coastal environment and are widespread globally. These bacteria make up a significant portion (up to 11%) of both the particle-attached and free-living bacterial communities in the coastal ocean, and they are present in every metagenome sample and at every time point that we have examined (Figs 1 and 2). We also found a greater variety of T6SS-bearing bacteria that exist mainly in the particle-attached fraction, including six genera seen exclusively in this type of habitat (Table 1). Only 4 of the 21 genera examined were primarily found in the free-living fraction. These patterns indicate that organic particles ranging from marine snow to living phytoplankton can also be potential hotspots for bacterial T6SS activity. Just like mucus in corals, live phytoplankton also secretes transparent exopolymeric substances that attract various heterotrophic bacteria. Interbacterial competition in a particulate is on a localized scale, and the encounter rate with other microbes is higher, as particles in the ocean tend to be hotspots of microbial activity [42, 43]. However, being particle-attached limits their prey selection to their immediate microbe neighbors, targeting competing bacteria with the same particle-attached lifestyle. Some T6SS-bearing bacteria also target eukaryotic host cells [44, 45], but it remains to be seen if T6SS is being used as a mechanism for algicidal activities. *Shewanella*, *Vibrio*, *Sphingomonas*, *Halomonas* and *Pseudo-alteromonas* genera, for example, are known for their algalytic tendencies toward dinoflagellates and raphidophytes [46–49]. On the other hand, T6SS-bearing bacteria are less diverse in the free-living fraction (4 out of 21) than particle-attached bacteria. Free-living bacteria tend to be highly dispersed in the water column; thus, potential competitors' encounter rate could be slim. Hence, having a T6SS could only become advantageous under particular conditions that can promote interbacterial competition.

Our study has shown that T6SS-bearing bacteria increase in abundance when fewer resources are available. This pattern is seen temporally in the SC Wharf samples and across large areas in the TARA Oceans samples. Notably, the abundance of some T6SS-bearing bacteria in SC Wharf samples was tied positively to nutrient concentrations, but many are negatively associated with diatom bloom and chlorophyll biomass. These suggest that other heterotrophic bacteria are putatively outnumbering T6SS-bearing bacteria during high phytoplankton biomass, which tend to sustain high bacterial production. During phytoplankton blooms and nutrient-rich upwelling events, bacteria can access large pools of organic carbon and other growth factors [50–52]. However, when upwelling and phytoplankton blooms cease, heterotrophic bacteria must compete over much more limited particulate and dissolved organic nutrients. Our time series data indicate that T6SS-bearing bacteria thrive during these times relative to high phytoplankton biomass seasons.

Global samples further reveal similar environmental trends in T6SS bacterial abundance, considering the sampling method's differences. The strongest effect seen in TARA Oceans data was increasing abundance of T6SS genes with depth, with a signal of decreasing abundance with increased autotroph abundance. The global patterns of T6SS gene frequency also indicate that the abundance of T6SS-bearing bacteria increases as resources become scarce. These bacteria were seen to be more prevalent in deeper water, as available particulate organic carbon (POC) and dissolved organic carbon (DOC) tend to diminish with depth in the euphotic zone [53, 54]. Moreover, global patterns indicate that these bacteria are more prevalent in areas with fewer primary producers, such as the gyres and non-coastal environment (Fig 4). Overall, these results agree with our local time-series results and demonstrate that the presence and abundance of T6SS-bearing bacteria occur at times and places where resources are limited. Under increased bacterial competition for limited resources, the evolution or acquisition of T6SS by heterotrophic bacteria may allow for the expansion of their ecological niche by capably attacking other bacteria to acquire extra nutrients or reducing local competition for the

same resources. In the colonization of the light organ crypts in bob-tail squid *Euprymna scolopes*, for instance, one lethal strain of *Vibrio fischeri* successfully eliminated a non-lethal strain of *V. fischeri* in occupying the squid's light organ crypt by using its T6SS [55]. Similarly, the T6SS mechanism is viewed as a molecular weapon and used in many cases to outcompete bacterial and eukaryotic competitors [56]. Thus, having a functional T6SS is likely an adaptation to temporal and spatial resource limitation and may explain the retention of this evolutionary trait in some bacterial genera.

Bacterial community structure is known to be influenced by top-down control such as protist grazers that feed on heterotrophic bacteria [9, 57], bottom-up control such as the availability of nutrients and organic matter [58, 59] or bacteria-bacteria allelopathy, such as the production of antibiotics [60]. However, the impact of T6SS-mediated interbacterial competition on bacterial assemblages has rarely been considered a potential factor. Although our study did not look at the environmental factors when T6SS is functionally expressed and actively employed by the examined bacterial genera, our study indicated that bacteria with an evolved machinery to kill other bacteria are common in the pelagic environment. These bacteria can potentially play a role in changing the composition of the microbial community, given the right environmental trigger for T6SS activity. Considering potential interactions of this type might eventually lead to a deeper understanding of bacterial dynamics in marine systems and the effects of these interactions on the bacterial community's collective functions in the recycling of nutrients and the microbial transformation of organic carbon.

## Supporting information

**S1 Table. T6SS core gene references and accessions.**
(XLSX)

**S2 Table. BioSample IDs of sequences submitted in SRA database.**
(XLSX)

**S3 Table. Environmental characteristics for the Santa Cruz Wharf dataset.**
(DOCX)

**S4 Table. Environmental variables used in the RDA analysis for the Tara Oceans dataset.**
(CSV)

## Acknowledgments

The authors thank Sanjin Mehic and Terrill Yazzie for their contributions to sample collection, and Jiunn Fong for his support throughout the experiments. We also acknowledge the support from Myers Trust to MWK and the TARA Oceans project for its open-source data policies that made worldwide metagenomics comparisons possible.

## Author Contributions

**Conceptualization:** Marilou P. Sison-Mangus.

**Data curation:** Michael W. Kempnich, Marilou P. Sison-Mangus.

**Formal analysis:** Michael W. Kempnich.

**Investigation:** Michael W. Kempnich.

**Methodology:** Michael W. Kempnich, Marilou P. Sison-Mangus.

**Project administration:** Marilou P. Sison-Mangus.

**Supervision:** Marilou P. Sison-Mangus.

**Validation:** Marilou P. Sison-Mangus.

**Visualization:** Michael W. Kempnich, Marilou P. Sison-Mangus.

**Writing – original draft:** Michael W. Kempnich.

**Writing – review & editing:** Marilou P. Sison-Mangus.

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
