## [Decision Letter · Decision Letter 0]

22 Sep 2020

PONE-D-20-23455

Presence and abundance of bacteria with the Type VI secretion system in a coastal environment and in the global oceans

PLOS ONE

Dear Dr. Sison-Mangus,

Thank you for submitting your manuscript to PLOS ONE. First, I want to apologize for the long delay of this reviewing, but we have been waiting for the responses of a second reviewer, and despite regular reminders, we did not receive them. You will find below the extensive reviewing on one referee. As you will see, this referee acknolwedges that you report important findings, but lists a number of recommendations to improve the analyses and the manuscript. I therefore encourage you to carefully address the referee's comments and invite you to submit your revised manuscript by Nov 06 2020 11:59PM. If you will need more time than this to complete your revisions, please reply to this message or contact the journal office at plosone@plos.org. Please include the following items when submitting your revised manuscript:

We look forward to receiving your revised manuscript.

Kind regards,

Eric Cascales

Academic Editor

PLOS ONE

Journal Requirements:

2. In your Methods section, please provide additional location information of the collection site, including geographic coordinates for the data set if available.

3. In your Methods section, please provide additional information regarding the permits you obtained for the work. Please ensure you have included the full name of the authority that approved the collection site access and, if no permits were required, a brief statement explaining why.

4. Please ensure that you refer to Figure 3 in your text as, if accepted, production will need this reference to link the reader to the figure.

5. Please include a caption for figure 3.

Reviewers' comments:

Reviewer's Responses to Questions

**Comments to the Author**

1. Is the manuscript technically sound, and do the data support the conclusions?

Reviewer #1: Partly

2. Has the statistical analysis been performed appropriately and rigorously? 

Reviewer #1: Yes

3. Have the authors made all data underlying the findings in their manuscript fully available?

Reviewer #1: Yes

4. Is the manuscript presented in an intelligible fashion and written in standard English?

Reviewer #1: Yes

5. Review Comments to the Author

Reviewer #1: In this study, the authors use bacterial DNA isolated from marine water samples from several locations and at different time points to analyze the relative abundance of genera carrying genes encoding structural components of the type VI secretion system (T6SS). One of their main findings is that T6SS-carrying bacteria appear to be more prevalent in the particle-attached fraction of their samples. They also correlate relative abundances with several environmental parameters, such as nutrient concentrations, temperature etc.

The study investigates the potential influence of an interesting bacterial phenotype (the T6SS machinery) on bacterial communities in natural environments, and I find the results to be intriguing and the manuscript well-written overall.

My only major complaints would be the following:

a) Relative abundances generally, and putative predation networks specifically. Fig. 6 shows interaction analyses based on relative abundance data of T6SS-containing bacteria in different samples and different time points (Santa Cruz wharf dataset). I am a bit sceptical of this analysis and the claims based on it, as these putative interaction networks are purely based on high-throughput sequencing data, and the relationship between absolute abundance in the environment (i.e. what we are actually interested in) and the relative abundance after sequencing (what we analyse) is often not predictable. See e.g. Fig. 1B and 1C here: https://www.frontiersin.org/articles/10.3389/fmicb.2017.02224/full

Given these inherent shortcomings of relative abundances as a metric, I am not convinced that the measured changes in relative abundances can be linked to an interaction between species here. If the analysis the authors did accounts for what is discussed in the paper above, I suggest this be addressed in this manuscript as this figure immediately raised a red flag in my mind. The same point of criticism applies to relative abundances as used in Fig. 2 in this study, and I’d be interested to hear the authors’ point of view on how their analysis might be generally influenced by the intrinsic shortcomings of relative abundance data.

b) Labelling bacterial genera containing T6SS-structural genes as “predatory” (throughout the manuscript). “Predation” in the strict sense implies not only killing, but also consumption of the targeted cells and nutritional exploitation (as the authors explain in the introduction). Although the T6SS is putatively involved in some bacterial predatory phenotypes sensu strictu (e.g. in Myxococcus), the T6SS is extremely wide-spread among Gram-negative bacteria and a role for it in actual “predation” has not been shown for any other species to the best of my knowledge. In the vast majority of species where the T6SS has been studied mechanistically in the lab in great detail, the T6SS is used as anti-competitor weaponry to kill competitors, where the main benefit is not nutrient uptake from the killed cell but rather just the physical removal of a competitor, which leads to more space for growth, and decreased nutrient-based competition. The chain of showing predation would need to include: find T6SS genes  show that T6SS is expressed  show that T6SS kills competitors  show that nutrients from killed cells are taken up and significantly enhance fitness of the attacker. Since the study at hand only focusses on finding T6SS genes, I suggest the label “predatory” bacteria is removed and replaced by a more conservative label, e.g. “T6SS-harbouring bacteria” or something along those lines. Similarly, instead of saying “predating on competitors” I would say “T6SS-dependent killing or -inhibiting of competitors”.

In summary, I think this study contains interesting results on the distribution of T6SS-genes in marine environments but could use with some more discussion on how representative relative abundances really are for environmental communities, and I would also suggest not labelling bacterial genera harbouring T6SS genes as “predatory bacteria”, as several aspects of this definition are not fulfilled by this study system.

Minor comments:

1. L82 -84: “Information is currently scarce …” We do have quite a bit of information on how the T6SS affects survival and success in host-associated communities though (e.g. Vibrio in squids https://www.pnas.org/content/115/36/E8528; Salmonella in guts https://pubmed.ncbi.nlm.nih.gov/27503894/; and others), which should fall under an “ecological setting” as the authors have phrased it here. We do indeed have less information on non-host-associated environments like the ocean, and if this is the gap that the authors would like to emphasize than I suggest to rephrase this statement accordingly.

2. L84 Do  does (T6SS most often written as singular, the T6SS as a whole)

3. L103 bacteria populations  bacterial populations

4. L103 bacteria-bacteria  Interbacterial (just a suggestion)

5. L104 in  for

6. L111 for  for the

7. L236 bacteria  bacterial

8. Fig. 2: I’m a bit confused: why are there gaps in the plot at certain time points? is it because no DNA was isolated here, no T6SS-containing genera were present here, or no samples were taken (but then why plot)?

9. Fig 2. Legend: “Error bars … across those samples” Sample size should be indicated in all figure legends for figures that contain errors bars.

10. L296 Fig. 4 is cited, but I think this should be Fig. 3 (the one with the global map)

11. L300 Fig. 4 Legend, but I think this should be Fig. 3 Legend

12. L339 p-value: throughout the study, are p-values corrected for multiple testing? If yes, this should be mentioned in the method section.

13. L349-352: add units for all variables/environmental factors

14. L337+L339 + L366 p-values are > 0.05, which is above the significance threshold of 0.05 used in this study (or at least above all the thresholds I saw explicitly mentioned, e.g. in legends for Fig. 5, Fig. 6, and in the methods section). P-value cut-offs are of course inherently arbitrary, but if different cut-offs were used for different analyses this should be mentioned somewhere (e.g. in methods section). If these values are indeed considered “non-significant”, the claims based on the respective comparisons should be amended in the main text.

15. L373-375: add units for all variables/environmental factors

16. Fig. 6 Legend – since two different datasets are used in this study (SC wharf vs. TARA) it would be good to indicate here which dataset this specific interaction analysis is based on.

17. Fig. 6 title – I suggest changing the figure title to “putative predation networks”, since “predation network” strongly suggests that pairwise interaction assays were conducted experimentally, with cultured isolates.

6. PLOS authors have the option to publish the peer review history of their article (what does this mean?). If published, this will include your full peer review and any attached files.

Reviewer #1: **Yes: **Elisa T. Granato

---

## [Author Response · Author response to Decision Letter 0]

25 Nov 2020

Journal Requirements:

 - we have read PLOS One requirements and adhered thoroughly to the instructions, especially for file naming. 

2. In your Methods section, please provide additional location information of the collection site, including geographic coordinates for the data set if available.

 - Coordinates have been added to the sampling site location information on Line 104

3. In your Methods section, please provide additional information regarding the permits you obtained for the work. Please ensure you have included the full name of the authority that approved the collection site access and, if no permits were required, a brief statement explaining why.

 -No permits were required for gathering samples, as Santa Cruz Wharf is open to the public. A statement to this affect has been added to the methods section in Line 105. 

4. Please ensure that you refer to Figure 3 in your text as, if accepted, production will need this reference to link the reader to the figure.

 - Previously, Figure 3 erroneously called it figure 4 in the text. This has been corrected to Fig 3. See line 327.

5. Please include a caption for figure 3.

 - We have provided a caption for corrected fig 3 (see above). See Line 328.

Comments to the Author

Reviewer #1: In this study, the authors use bacterial DNA isolated from marine water samples from several locations and at different time points to analyze the relative abundance of genera carrying genes encoding structural components of the type VI secretion system (T6SS). One of their main findings is that T6SS-carrying bacteria appear to be more prevalent in the particle-attached fraction of their samples. They also correlate relative abundances with several environmental parameters, such as nutrient concentrations, temperature etc.

The study investigates the potential influence of an interesting bacterial phenotype (the T6SS machinery) on bacterial communities in natural environments, and I find the results to be intriguing and the manuscript well-written overall.

My only major complaints would be the following:

a) Relative abundances generally, and putative predation networks specifically. Fig. 6 shows interaction analyses based on relative abundance data of T6SS-containing bacteria in different samples and different time points (Santa Cruz wharf dataset). I am a bit sceptical of this analysis and the claims based on it, as these putative interaction networks are purely based on high-throughput sequencing data, and the relationship between absolute abundance in the environment (i.e. what we are actually interested in) and the relative abundance after sequencing (what we analyse) is often not predictable. See e.g. Fig. 1B and 1C here: https://www.frontiersin.org/articles/10.3389/fmicb.2017.02224/full

 - We do agree that HTS 16S-sequences are compositional data and do not reflect the exact number of molecules of bacteria in a particular sample. Hence relative abundances have inherent limitations and may not be suitable dataset to studying correlations. We followed the advice of Gloor et al. 2017 paper (as pointed out by the reviewer) and have revised the analysis using centered log-ratio transformations which uses the geometric mean of the sample vector as the reference, a CoDA analysis framework formulated by Aitchison 1986. We followed the methods laid out in Quinn et al. 2019 and have described the detailed analysis in the Materials and Methods Line 171- 191.

We carried out the correlational analysis between environmental variables and populations of T6SS-bearing bacteria after clr transformations using Redundancy Analysis (RDA), a variant of PCA, as the data are now symmetrical and linearly related and amenable for correlational analysis. Details of the statistical methods were described under Materials and Methods Line 203-223.

Given these inherent shortcomings of relative abundances as a metric, I am not convinced that the measured changes in relative abundances can be linked to an interaction between species here. If the analysis the authors did accounts for what is discussed in the paper above, I suggest this be addressed in this manuscript as this figure immediately raised a red flag in my mind. 

 - The interactions derive from figure 6 were putative interactions, where hypothesis was formulated to guide actual, live experiments of the interaction between T6SS-bearing bacteria and other bacterial groups. Even though that is the intent of the graph, we agree that the nature of the data (based on sequencing and relative abundance) may not be robust to produce those putative hypotheses. We have removed Figure 6 in the revised manuscript in this regard, and even in doing so, it did not diminish the main message of the paper. 

The same point of criticism applies to relative abundances as used in Fig. 2 in this study, and I’d be interested to hear the authors’ point of view on how their analysis might be generally influenced by the intrinsic shortcomings of relative abundance data.

 -As for Fig 2, we re-did the analysis by doing centered log ratio transformation and the transformed data was plotted as a heatmap to visualized how the OTUs have changed in abundance relative to the rest of the bacterial community (which is using geometric mean as a reference, based on compositional analysis by Aitchison 1986). By doing this, we only found a slight change in the result and interpretation of the data, and we have made this modification accordingly in the text. See Lines 260 – 275.

b) Labelling bacterial genera containing T6SS-structural genes as “predatory” (throughout the manuscript). “Predation” in the strict sense implies not only killing, but also consumption of the targeted cells and nutritional exploitation (as the authors explain in the introduction). Although the T6SS is putatively involved in some bacterial predatory phenotypes sensu strictu (e.g. in Myxococcus), the T6SS is extremely wide-spread among Gram-negative bacteria and a role for it in actual “predation” has not been shown for any other species to the best of my knowledge. In the vast majority of species where the T6SS has been studied mechanistically in the lab in great detail, the T6SS is used as anti-competitor weaponry to kill competitors, where the main benefit is not nutrient uptake from the killed cell but rather just the physical removal of a competitor, which leads to more space for growth, and decreased nutrient-based competition. The chain of showing predation would need to include: find T6SS genes  show that T6SS is expressed  show that T6SS kills competitors  show that nutrients from killed cells are taken up and significantly enhance fitness of the attacker. Since the study at hand only focusses on finding T6SS genes, I suggest the label “predatory” bacteria is removed and replaced by a more conservative label, e.g. “T6SS-harbouring bacteria” or something along those lines. Similarly, instead of saying “predating on competitors” I would say “T6SS-dependent killing or -inhibiting of competitors”.

 -We have removed the term predatory bacteria in the revised texts and used T6SS- bearing bacteria as suggested. We also used the term interbacterial competition to refer to the interaction. 

In summary, I think this study contains interesting results on the distribution of T6SS-genes in marine environments but could use with some more discussion on how representative relative abundances really are for environmental communities, and I would also suggest not labelling bacterial genera harbouring T6SS genes as “predatory bacteria”, as several aspects of this definition are not fulfilled by this study system.

 - We followed this suggestion of the reviewer and emphasized that our study focuses on the T6SS-bearing heterotrophic bacteria living in the marine water column or the marine pelagic environment, which to our knowledge, has rarely been addressed in the literature. 

Minor comments:

1. L82 -84: “Information is currently scarce …” We do have quite a bit of information on how the T6SS affects survival and success in host-associated communities though (e.g. Vibrio in squids https://www.pnas.org/content/115/36/E8528; Salmonella in guts https://pubmed.ncbi.nlm.nih.gov/27503894/; and others), which should fall under an “ecological setting” as the authors have phrased it here. We do indeed have less information on non-host-associated environments like the ocean, and if this is the gap that the authors would like to emphasize than I suggest to rephrase this statement accordingly.

 - As stated above, we have revised the paper and emphasized that our study focuses on T6SS-bearing bacteria living in the pelagic environment. For specifics, see lines 16-21 in the abstract, Line 79-82 in the introduction and have subsequently discussed the results and implications throughout the Discussion section. 

2. L84 Do  does (T6SS most often written as singular, the T6SS as a whole)

 - This sentence or question has been removed in the introduction. As the reviewer suggested, sequence data may not be able to address this type of question, hence the removal of the statement/question. 

3. L103 bacteria populations  bacterial populations

 -This has been corrected. See Line 93. 

4. L103 bacteria-bacteria  Interbacterial (just a suggestion)

 - We have removed the sentence containing this hyphenated word in Line 94 (previously L103) but we have used this suggested wording throughout the paper when we talk about bacteria-bacteria competition. For example, see Line 78, 80, 441 etc.

5. L104 in  for

 - This has been corrected. See Line 94.

6. L111 for  for the

 - This has been corrected. See Line 100. 

7. L236 bacteria  bacterial

 - This has been corrected. See Line 256.

8. Fig. 2: I’m a bit confused: why are there gaps in the plot at certain time points? is it because no DNA was isolated here, no T6SS-containing genera were present here, or no samples were taken (but then why plot)?

 - The reviewer is correct, there were no samples taken at these time points. These empty datapoints were removed. We subsequently changed the graph into a heatmap figure to better show the differences in the occurrence of the T6SS-bearing bacteria through time. The data was treated as compositional dataset and were subjected to centered log-ratio transformation, which take into account the abundance changes of T6SS-bearing bacteria using the geometric mean per sample as reference. See revised Fig. 2.

9. Fig 2. Legend: “Error bars … across those samples” Sample size should be indicated in all figure legends for figures that contain errors bars. 

 - As mentioned above, the figure has been changed into a heatmap and all replicates for each sample were included in the figure. We have also indicated the sample size (duplicate and triplicate) in the figure legends. Please see legend in Fig 2 Line 283-284.

10. L296 Fig. 4 is cited, but I think this should be Fig. 3 (the one with the global map)

 - Thank you for pointing out the mistake. We have corrected this as Fig. 3.

11. L300 Fig. 4 Legend, but I think this should be Fig. 3 Legend

 -Please see response above. The legend now says Fig 3. See Line 329.

12. L339 p-value: throughout the study, are p-values corrected for multiple testing? If yes, this should be mentioned in the method section.

 -We have revised the analysis and used Benjamini-Hochberg correction for multiple testing and used the resultant p-values. See Line 196 in Materials and method sections, where we mentioned the method. 

13. L349-352: add units for all variables/environmental factors

 - We have added the units in the text of Fig 4 legend as suggested by the reviewer (see Lines 381), as well as in other places where they occur, such as in the results section/legends.

14. L337+L339 + L366 p-values are > 0.05, which is above the significance threshold of 0.05 used in this study (or at least above all the thresholds I saw explicitly mentioned, e.g. in legends for Fig. 5, Fig. 6, and in the methods section). P-value cut-offs are of course inherently arbitrary, but if different cut-offs were used for different analyses this should be mentioned somewhere (e.g. in methods section). If these values are indeed considered “non-significant”, the claims based on the respective comparisons should be amended in the main text.

 - As the author mentioned, p-value cut-offs are arbitrary. As for any statistical analysis, we use the p-value cut-off of 0.05 (after correcting for multiple testing), when indicating a specific environmental variable as having a significant contribution to the structure of T6SS-bearing populations. As mentioned above, we re-did the analysis after clr transformation of the dataset using Redundancy analysis, which gave a robust multivariate correlation between the environmental determinants and T6SS-bearing bacteria. See Lines 343-413. 

15. L373-375: add units for all variables/environmental factors

- Units have been added to the text in Fig. 5 legend. See Lines 41 to 424. We have also added the units in the results section/legends where they occur. 

16. Fig. 6 Legend – since two different datasets are used in this study (SC wharf vs. TARA) it would be good to indicate here which dataset this specific interaction analysis is based on.

 - Figure 6 has been removed from the revised paper. 

17. Fig. 6 title – I suggest changing the figure title to “putative predation networks”, since “predation network” strongly suggests that pairwise interaction assays were conducted experimentally, with cultured isolates.

 - Figure 6 has been removed from the revised paper.

---

## [Decision Letter · Decision Letter 1]

7 Dec 2020

Presence and abundance of bacteria with the Type VI secretion system in a coastal environment and in the global oceans

PONE-D-20-23455R1

Dear Dr. Sison-Mangus,

Thank you for submitting your revised manuscript, and for having addressed the comments of the reviewer. Your revised manuscript has been sent back to the original reviewer, who acknowledges that you have appropriately addressed her/his intiial comments and recommends publication. I am therefore very pleased to accept your manuscript. Please note that your work will be formally accepted for publication once it meets all outstanding technical requirements.

Kind regards,

Eric Cascales

Academic Editor

PLOS ONE

Additional Editor Comments (optional):

Reviewers' comments:

Reviewer's Responses to Questions

**Comments to the Author**

1. If the authors have adequately addressed your comments raised in a previous round of review and you feel that this manuscript is now acceptable for publication, you may indicate that here to bypass the “Comments to the Author” section, enter your conflict of interest statement in the “Confidential to Editor” section, and submit your "Accept" recommendation.

Reviewer #1: All comments have been addressed

2. Is the manuscript technically sound, and do the data support the conclusions?

Reviewer #1: (No Response)

3. Has the statistical analysis been performed appropriately and rigorously? 

Reviewer #1: (No Response)

4. Have the authors made all data underlying the findings in their manuscript fully available?

Reviewer #1: (No Response)

5. Is the manuscript presented in an intelligible fashion and written in standard English?

Reviewer #1: (No Response)

6. Review Comments to the Author

Reviewer #1: (No Response)

7. PLOS authors have the option to publish the peer review history of their article (what does this mean?). If published, this will include your full peer review and any attached files.

Reviewer #1: No

---

## [Editor Report · Acceptance letter]

10 Dec 2020

PONE-D-20-23455R1 

Presence and abundance of bacteria with the Type VI secretion system in a coastal environment and in the global oceans 

Dear Dr. Sison-Mangus:

I'm pleased to inform you that your manuscript has been deemed suitable for publication in PLOS ONE. Congratulations! Your manuscript is now with our production department. 

Kind regards, 

on behalf of

Dr. Eric Cascales 

Academic Editor

PLOS ONE